# Allosteric regulation in STAT3 interdomains is mediated by a rigid core: SH2 domain regulation by CCD in D170A variant

**Tingting Zhao**, **Nischal Karki, Brian D. Zoltowski, Devin A. Matthews**\*

Department of Chemistry, Southern Methodist University, Dallas, Texas, United States of America

\* dmatthews@smu.edu

**Data Availability Statement:** All relevant data are within the manuscript and its Supporting information files. Starting structures used for simulations, the raw data of the trajectories files

## Abstract

Signal Transducer and Activator of Transcription 3 (STAT3) plays a crucial role in cancer development and thus is a viable target for cancer treatment. STAT3 functions as a dimer mediated by phosphorylation of the SRC-homology 2 (SH2) domain, a key target for therapeutic drugs. While great efforts have been employed towards the development of compounds that directly target the SH2 domain, no compound has yet been approved by the FDA due to a lack of specificity and pharmacologic efficacy. Studies have shown that allosteric regulation of SH2 via the coiled-coil domain (CCD) is an alternative drug design strategy. Several CCD effectors have been shown to modulate SH2 binding and affinity, and at the time of writing at least one drug candidate has entered phase I clinical trials. However, the mechanism for SH2 regulation via CCD is poorly understood. Here, we investigate structural and dynamic features of STAT3 and compare the wild type to the reduced function variant D170A in order to delineate mechanistic differences and propose allosteric pathways. Molecular dynamics simulations were employed to explore conformational space of STAT3 and the variant, followed by structural, conformation, and dynamic analysis. The trajectories explored show distinctive conformational changes in the SH2 domain for the D170A variant, indicating long range allosteric effects. Multiple analyses provide evidence for long range communication pathways between the two STAT3 domains, which seem to be mediated by a rigid core which connects the CCD and SH2 domains via the linker domain (LD) and transmits conformational changes through a network of short-range interactions. The proposed allosteric mechanism provides new insight into the understanding of intramolecular signaling in STAT3 and potential pharmaceutical control of STAT3 specificity and activity.

## Author summary

In all living organisms, the proliferation and survival of cells are regulated by various proteins. Signal Transducers and Activators of Transcription 3 (STAT3) protein is one of these important proteins. However, the abnormal regulation of these proteins will contribute to the proliferation of cancer. The constitutive activation of STAT3 has been linked to several types of solid tumors, leukemia, and lymphomas. Consequently, STAT3

and pymol files are avaliable at https://osf.io/dvzq7/.

**Funding:** The authors acknowledge funding sources, including NSF research grant No. (1753167) https://www.nsf.gov/. The funders had no role in study design, data collection and analysis, decision to publish, or preparation of the manuscript.

**Competing interests:** The authors have declared that no competing interests exist.

proteins have been a key target for cancer therapy. SH2 (SRC-homology 2) domain is the key interaction site, great efforts have been made to target SH2 domain. However, specificity has been a major challenge in drug discovery. Research showing regulation of SH2 domain via CCD (coiled-coil domain) has opened a new path for drug discovery, but progress is challenged by poor understanding of the allosteric mechanism. Here, we show that CCD regulates SH2 conformation via a rigid backbone. The perturbations in CCD are transmitted through an $\alpha$-helix to the rigid core that orchestrate the movement of CCD and LD (link domain), leading to structural changes in the SH2 domain. The present findings provide an allosteric mechanism with atomistic details underlying the regulation of CCD to SH2 domain in STAT3 protein. A detailed allosteric pathway allows informed drug design targeting CCD for desired downstream effect on SH2 domain and the overall STAT3 function.

## Introduction

Proteins within the Signal Transducers and Activators of Transcription (STAT) family function as both signal transducers in the cytoplasm and transcription factors upon nuclear translocation. All members of STAT family consists of six domains (Fig 1A): amino-terminal domain (NTD), coiled-coil domain (CCD), DNA-binding domain (DBD), linker domain (LD), SRC-homology 2 domain (SH2), and transactivation domain (TAD) which is also named the C terminal domain [1]. STAT proteins are regulated by Janus Kinases (JAKs)

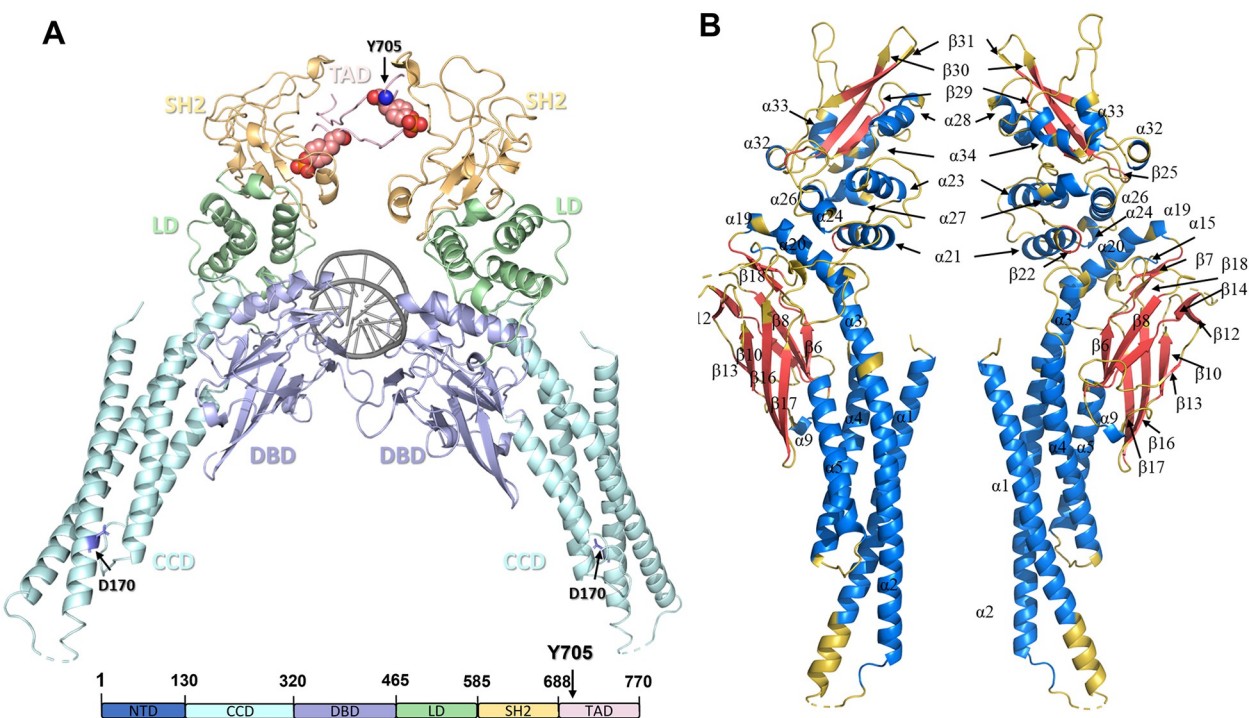

**Fig 1. STAT3 structure.** 1BG1 was used as the template; NTD is not shown. **(A)** STAT3 domain structure (Y705 is shown as spheres, D170 is shown as sticks). Vide infra for details of the initial structure.**(B)** Secondary structures are labeled according to the UniProt database (S1 Table) [6]: $\alpha$ helices are colored blue, $\beta$ sheets are colored red, and unstructured regions (loops) are colored yellow (transverse view). The assigned secondary structures combine information from multiple x-ray crystal structures, thus there is some mismatch with the specific structures used in this work.

where they play a crucial role in immune response, cell division, and apoptosis, as a gene expression regulatory arm of JAK-STAT signaling pathway [2]. However, each member is activated via different types of cytokines and have unique function in the pathway [3–5].

Constitutive activation of STAT3 has been shown to play a crucial role in cancer progression [7, 8]. STAT3 binds, via the SH2 domain, to cell-surface receptors upon activation and recruitment of receptor-associated kinases. Upon binding, the recruited kinases activate STAT3 through phosphorylation within the TAD (at Y705), followed by dissociation from the receptor to form homodimers through reciprocal interactions between the SH2 domain and the phosphotyrosine (pY705) residue. These activated homodimers are then translocated to the nucleus where the DBD binds to target genes, and TAD activates the expression of proteins crucial for cell growth and survival. In normal cells, the signaling pathway is well-regulated. However, the abnormal activation of this signaling pathway promotes the development of cancer: misregulation of STAT3 in cancer cells promotes pro-oncogenic inflammation and suppresses anti-tumor immunity [9].

The direct therapeutic inhibition of STAT3 is highly desirable but remains challenging as evident from the lack of FDA–approved drugs. Specifically, significant amounts of effort have been employed to develop molecules targeting the SH2 domain of STAT3 [10–12]. The SH2 domain is a structurally conserved protein domain, which appears in many intracellular signal transducing proteins, offering a binding site for phosphorylated tyrosine residues. The SH2 domain in STAT3 contains two regions with specialized functions: the pY pocket, into which the phosphotyrosine of the target inserts, is the binding region, while residues of the pY+3 pocket interact with the three C-terminal residues of the phosphotyrosine in the target, forming a specificity–determining region [13, 14]. The inhibitors targeting the SH2 domain: phosphotyrosine motifs (pY-peptide) or phosphotyrosine-based peptidomimetic inhibitors which mimic the pTyr-Xaa-Yaa-Gln motif, have been previously investigated, as well as the associated binding mode [15–17]. In the pY pocket, R609 is the principal binding partner, along with K591, S636 and S611 which directly interact with pY705. The relative conformation and position of these residues will have a direct effect on STAT3 binding activity. In the pY+3 pocket, V637 in $\beta$31 controls accessibility to this pocket, while Y657, Q644, Y640, and E638 facilitate the hydrogen bond interaction with its target, as well as I659, W623 and F621 assist in binding of target peptide by forming hydrophobic environment [17]. However, most of these compounds have yet to be explored in clinical studies or further development of these compounds was limited due to concerns with their relative lack of potency and selectivity [18].

Several studies [20–22] have determined that effectors (small molecule and polypeptide) binding to CCD interfere with SH2 domain binding or preclude STAT3 nuclear translocation, which suggests CCD as a potential target for further drug design. Zhang et al. found that the coiled-coil domain is essential for STAT3 recruitment to the receptor: systematic deletion analysis of the N-domain and $\alpha$ helices of CCD, as well as mutagenesis of conserved residues (D170A) in the CCD of STAT3 were carried out and showed the diminishment of both pY-peptide binding and tyrosine phosphorylation [19]. Furthermore, the small molecule MM-206 was identified as an inhibitor of STAT3 phosphorylation, and Minus et al. surprisingly found the binding site was at $\alpha$1 of CCD (around F174). In addition, the compound K116, found to bind to CCD by AlloFinder, was shown to be able to inhibit receptor binding, validated by mutagenesis and functional experiments [21]. Recently, a small polypeptide MS3–6 was found to bind to CCD, which caused significant helical tilt in CCD domain compared to the apo conformation, which further diminished DNA binding and nuclear translocation [22]. These observations are summarized in Table 1, and highlight CCD correlation to SH2 domain binding affinity as well as specificity.

**Table 1. Summary of STAT3 effectors targeting CCD and their effect on pY-peptide and DNA binding activity.** a) [19], b) [20], c) [21], d) [22].

| Alteration/Effector | pY-Peptide Binding | | DNA Binding |
| --- | --- | --- | --- |
| | Sequence / Activity(SPR-based assay) | Phosphorylation Agent /Activity (western blot) | |
| D170A mutation[a] | VVHSG(pY)RHQVPS / Inhibited | EGF / Inhibited | Not studied |
| MM-206[b] | LPVPE(pY)INQSVP / Inhibited | IL-6 / Inhibited | Inhibited[b] |
| K116[c] | Ac-pYLPQTV-NH2 / Inhibited | IL-6 induced / Inhibited | Not studied |
| MS3–6[d] | GMPKS(pY)LPQTVR / Not inhibited | IL-22 / Inhibited<br>IL-6 / Not inhibited | Diminished |

The discovery of several and diverse inhibitory agents, which bind to the CCD rather than SH2, but regulate SH2 domain function, is a fascinating development. However, rational design of allosteric effectors requires a more detailed, mechanistic knowledge of how CCD binding affects SH2 structure and activity. There are no crystal structures with MM-206 or K116 binding with STAT3. Moreover, computational modeling of large fragments and optimizing the accurate binding configuration is challenging. D170A mutation, which has shown altered activity in SH2 domain, can thus be a useful avenue of investigation towards understanding the allosteric mechanism. This has the further advantage in that a single point mutation can easily be performed computationally without significantly altering the structural environment. We hypothesize that point mutation and effector binding result in a similar allosteric pathway outside of the CCD domain (where local interactions dominate). While we do not explicitly test this hypothesis here, unraveling the D170A allosteric mechanism will at least help to guide the search for allosteric mechanisms of alternate effectors.

In this work, we study the allosteric mechanism of D170A mutation on the inhibition of STAT3 activity, as predicted by the dynamic structures of the SH2 domain, known to be essential for pY-peptide binding and ensuing Y705 phosphorylation. Specifically, we investigate the structural properties within the SH2 domain upon CCD mutation over a number of molecular dynamics simulations, as well as the dynamical correlations between SH2 and CCD and the associated networks of allosteric residues and interactions within various structural motifs.

## Results

### Conformational differences in the pY+3 binding pocket correlate to the decreased binding affinity from wild type to D170A variant

The SH2 domain mediates binding of kinase-complexes to unphosphorylated STAT3, directing the phosphorylation of Y705 at TAD. Furthermore, the SH2 domain also provides the interface for dimerization of the phosphorylated TAD to form a functional pSTAT3 homodimer. Thus, any changes to the conformation of the specificity-determining region (pY+3 pocket, Fig 2D) as well as the binding region of phosphorylated TAD (pY pocket, Fig 2D) provides a key regulatory modification to STAT3 behavior. To describe the binding pocket conformations of SH2 domain, the pair residue center of mass (COM) distance matrix of key residues (Fig 2D, see Methods section for details) for both the pY and pY+3 pockets were calculated, separately. Principal component analysis (PCA) was employed to project the high dimensionality of pair residue COM distances into a 2D plane. The conformational space of both pockets for wild type and D170A variant is shown in Fig 2A (note that the principal components are determined from the combined wild type and D170A variant trajectories). The first two principal components contribute 62% of the total variation (S1(A) Fig), encapsulating the majority of the conformational space in just two dimensions.

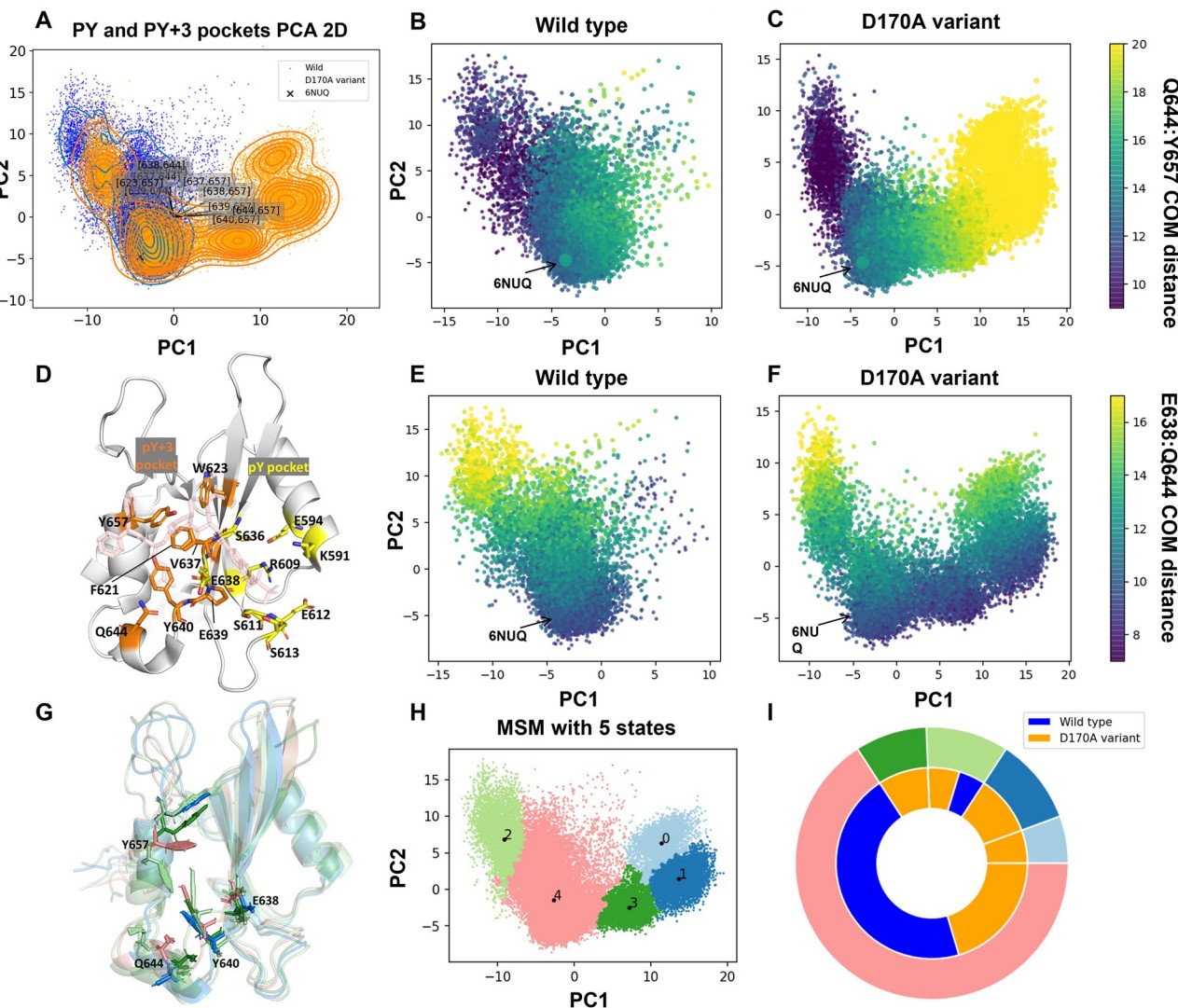

**Fig 2. Conformational analysis of SH2 domain binding pockets. (A)** PCA of both the pY and pY+3 pockets in the wild type (blue) and D170A mutant (orange). The contour lines show the density of recorded frames in each region, and the crystal structure (PDB ID: 6NUQ) is marked for reference. 6NUQ is STAT3 with a ligand bound to the SH2 domain, shown as a reference for the ligand binding conformation. **(D)** SH2 domain bound to SI109 (light pink) to demonstrate binding mode of ligand to the pY and pY+3 pocket (6NUQ [11]). The pY pocket (yellow) and pY+3 pocket (orange) are shown with key residues included in the COM pair distances are shown as sticks. **(B,C)** PCAs for wild type and D170A variant, respectively, colored by the COM distance from Q644 to Y657. **(E,F)** PCAs for wild type and D170A variant, respectively, colored by the COM distance from Q644 to E638. The distance between 644—657 and 638–644 can be used as an approximate representation of PC1 and PC2 (S1(C) Fig). The plot of these two distances distribution in the 12 simulations were shown in (S2 Fig). **(G)** The averaged structures for each macro-state with key residues show as sticks. **(H)** pY and pY+3 pockets PCA 2D plane colored by different macro-states. **(I)** Nested pie charts showing the degree to which each system (apo and D170A) occupy each macro-state. The outer circle is colored by macro-state as in (**H**), and the inner circle is colored by system.

In the combined PCA plot, the conformational space of the D170A variant partially overlaps with the wild type, while also exploring distinct novel conformations (Fig 2A). The additional conformational space explored by the D170A variant occurs predominately along the PC1 axis. The ten largest coefficients of the first two PCs highlight the pair of residues with largest relative motions across all the trajectories (S1(C) Fig). The motion of Y657 and Q644 yields the largest coefficients across PC1 and PC2 (Fig 2A), respectively, underlining their

conformational importance. The Q644–Y657 pair has the largest variance along PC1(Fig 2B and 2C) while the E638–Q644 pair has the largest variance along PC2 (Fig 2E and 2F).

Comparing Fig 2B, 2C, 2E and 2F with Fig 2A, we see that the E638–Q644 pair has a slightly higher variance in the wild type whereas the Q644–Y657 motion, particularly above 15 Å, is predominately a feature of the D170A variant. Additionally, the D170A variant explores somewhat shorter Q644–Y657 distances than in the wild type. Most notably, all of the residues with highest variance occur in pY+3 pocket rather than the pY pocket. This result is consistent with observation by Zhang et. al., where the authors show that D170A reduces binding affinity of ligands targeting the SH2 domain. The binding mode analysis performed by Dhanik et. al. shows that the binding affinity of a ligand is directly correlated to additional interactions in the pY+3 pocket [15]. In biomolecular recognition, the two aspects binding affinity and binding specificity are coupled to each other, i.e. strong binding affinity is indicative of high substrate specificity and vice versa. Furthermore, strong coupling among flexibility and binding affinity has been shown in different systems [23, 24]. Thus, higher flexibility would lead to exploration of extended conformational landscape and reduced occupancy in substrate binding conformation. The changes in flexibility ofpY+3 pocket could potentially explain the reduction of binding affinity observed by Zhang et. al.

To further investigate key differences in the overall structure, a combination of Markov State Modeling and Perron Cluster Cluster Analysis was applied to cluster transient conformations into kinetically meta-stable macro-states. Clustering into five macro-states (0–4) was applied to the combined wild type and mutant conformations (Fig 2H).

Both the D170A variant and wild type are well-represented within macro-states 2 and 4, while macro-states 0, 1 and 3 were uniquely explored by the D170A variant (Fig 2I). The average structures of each macro-state were calculated and are shown in Fig 2G. Distinct conformations of Q644 and Y657 are observed for each of the macro-states, in agreement with the PCA data. The pY+3 pocket is blocked in macro-state 2 by Y657 and Y640, both of them pointing towards the pocket. Conversely, in macro-state 0, 1, and 3, Y657 points away from the pY+3 pocket leading to an open conformation. The shared conformational states between wild type and D170A variant consists of a viable functional state of STAT3, however, D170A variant has reduced occupancy at those conformational states (Fig 2I), thus leading to a differentiated function relative to the wild type.

## Allosteric regulation of SH2 pY+3 occurs via translation of motion through a rigid core

To explore the correlation between SH2 domain conformational changes and CCD conformations, CCD was first characterized via PCA analysis of all pair Cα distances. However, as the CCD domain consists of rigid helices, pair Cα distances were not able to characterize the differences between the determined macro-states, as indicated by the low variance contribution of each pair Cα distance (S3(A), S3(C) and S3(D) Fig). Only macro-state 3 showed significant differences, which originate from kinked conformations of α1 (S3(B) Fig).

The rigidity of the α helices diminishes the utility of PCA withing the CCD alone to study intradomain conformational differences, however the communication between CCD and SH2 domains has been observed in the literature. To investigate this correlation, we hypothesize that the long helical arms of CCD may act as rigid levers, where any perturbation of the helical arm causes significant changes in inter-domain interaction sites. A role in the helical arm affecting the SH2 domain was in part motivated by prior studies demonstrating that monobodies targeting STAT3 interact at the helical arm and impact function through subtle bending and or rotation of helices in the CCD [22]. The argument for the subtle bending and or

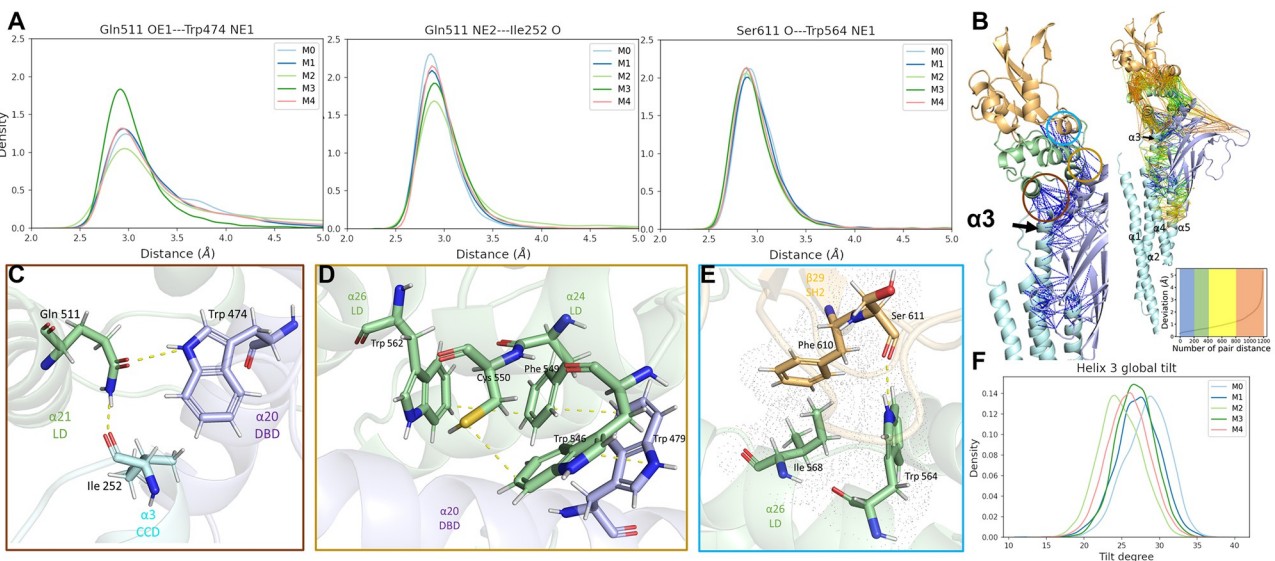

**Fig 3. Rigid core analysis showing interactions which are conserved across all macro-states.** The analysis includes all trajectories spanning macro-states M0–M4. **(A)** Distance distributions of the atom pair, that form hydrogen bonds, show similar distribution profiles. **(B)**Right: Inter-domain pair Cα distances (dashed lines) shows the intensity of coupling in the Cα pair motions between different domains. The pair Cα distances are colored from blue, green, yellow, and orange based on their deviation throughout the trajectory (Map in subplot). Left: A close-up of multi-domain interaction site shows the least first 200 deviating Cα pairs, demonstrating strongly correlated movements among CCD, LD and SH2 domain. **(C)** Conserved hydrogen bond network between CCD, LD, and DBD. **(D)** Conserved π-π interaction network between LD and DBD. **(E)** Conserved hydrogen bond network as well as hydrophobic interactions between LD and SH2. **(F)** Distribution of global helical tilt of α3 in CCD for each macro-state (M0–M4, see also supplemental S6 Fig). Statistical tests (T-test and Kolmogorov-Smirnov test) are shown in Table A in S1 Text

rotation of the helical arm impacting signal transduction is based on crystal structures where there are packing interactions that impact the helices. Thus, whether the movements of the helical arm are a result of the monobodies or crystal packing interactions remains an open point, and the MD simulation is of great value to follow up the crystallography studies. The hypothesis was tested by comparing corresponding helical tilt at the CCD of different macro-states identified in section 'Conformational differences in the pY+3 binding pocket correlate to the decreased binding affinity from wild type to D170A variant for the pY and pY+3 pockets'. Differences in global helical tilt for each of the macro-states of the SH2 domain is observed in α3 (Fig 3F). Although a subtle rotation, these are consistent with the helical movements found in the CCD due to MS3–6 binding that impact biological function. Further, given that α3 helix interfaces with most of the other domains through its C-terminal helical turn, it lies in a central position that may be able to transmit slight movements within the helical arm through subtle motions of interacting residues between α3 and nearby domains. Thus, although the helical tilt is small, and may not be a primary signaling mechanism, it may reflect movements of key residues at the interface between α3 and nearby domains that propagate conformational information from the CCD to the SH2 domain.

**Conserved Cα pair distances.** The rigidity transmission, where a perturbation of rigidity at one binding site can be allosterically transmitted to a second distant site, was observed in other proteins [25, 26]. Thus we further hypothesized that the motion of α3 is transmitted allosterically to SH2 via a "rigid core", that is, an interlocking sequence of conserved interactions which function as a sort of molecular machine. The existence of such an interaction network is demonstrated in Fig 3B, where the inter-domain pair Cα distances (See Methods) are plotted and colored according to standard deviation values computed across all trajectories. There is a rigid backbone through the protein from CCD, LD, DBD, and finally to SH2 (Fig 3B) which is

highly conserved during dynamical motion of the protein before and after mutation at D170. $\alpha$3, $\alpha$20, $\alpha$21 compose the first section of the rigid core between CCD, DBD, and LD (Fig 3C), which could convey the dynamics of CCD into this highly rigid region. Upon close inspection, the three helices $\alpha$3, $\alpha$20, and $\alpha$21 are locked via hydrogen bond network between I252, Q511 and W474 (Fig 3A and 3C), such that any rotation of these residues results in a corresponding reorientation of the helices to preserve the hydrogen bond network. We additionally find strongly conserved inter-domain interactions between the DBD and LD (Fig 3D), as well as between the LD and SH2 (Fig 3E), which complete the rigid core. Upon PCA analysis of residues of the rigid core, we see a minimal variance within the rigid core (S4(A) and S4(B) Fig). The PCA 2D plot shows two distinct conformational states, however, this distinction is primarily attributed to the pair C distances between residue 562 and residues from $\alpha$3 helix (S4 (C) Fig). Exclusion of the residue 562 from the PCA analysis of the rigid core shows an indistinguishable conformation landscape (S5(A) and S5(B) Fig). These factors allow subtle changes in the CCD configurations to convey movement from $\alpha$3 through $\alpha$21 (in LD) and $\alpha$20 (in DBD) to $\alpha$26 and $\alpha$24 (in LD), and finally leading to allosteric modification of SH2 via $\beta$29.

The conformation analysis presented above demonstrates that D170A mutation leads to changes in the orientation of the CCD $\alpha$ helices, which could then lead to allosteric regulation of SH2 domain conformations though a network of conserved interactions. Rigid body analysis shows that these interactions consist of hydrogen bond, $\pi$-$\pi$, and hydrophobic networks that strongly correlate motions of different domains. However, this analysis does not highlight the allosteric path that differs between the wild type and D170A variant nor provide evidence of a dynamical correlation between CCD and SH2 conformation through this pathway. To further elucidate the allostery pathway and show dynamical correlation, we employ both a REDAN analysis and analysis of differences in the global hydrogen bond network between kinetic macro-states.

**REDAN.** While a rigid backbone provides a potential pathway for signal propagation, the cumulative long-range allosteric effect is realized through short range interactions and subtle allosteric changes which must occur in concert. To identify a detailed sequence of short range interactions, REDAN analysis was employed as a means to identify residue pairs that are responsive to allosteric perturbation, followed by shortest path analysis using Dijkstra's algorithm. Using REDAN, subtle yet highly correlated differences in the allosteric network between D170A and the wild type can be resolved allowing us to propose a concrete signal transduction network from CCD to SH2.

From the conformational analysis, Y657 was identified as the residue with highest average difference between different macro-states explored by the SH2 domain. Thus, D/A170 was selected as the starting point and Y657 was selected as the end point for the analysis. The most structurally-relevant pathway from effector residue (D/A170) to regulatory site (Y657) was identified by REDAN and is shown in Fig 4A and 4B. The pathway originates from the CCD, through the LD and to the SH2 domain, bypassing DBD (although it passes nearby $\alpha$20 which was identified as a component of the rigid core). The residue pairs that connect domains are of most interest, and their distance distributions are shown in S7(C)–S7(E) Fig The average structures of each macro-state were calculated to structurally verify correlated motions in secondary structure identified by the REDAN (Fig 4C).

The secondary structure designated $\beta$22 (Fig 1B) can be observed to dynamically shift between a $\beta$ sheet and $\alpha$ helix in macro-states 2, 3, and 4, while in macro-state 0 and 1 the $\alpha$ helix is stabilized (S7(F) Fig). $\beta$22 extends the $\alpha$21 helix and reorganizes the loop between $\beta$22-$\alpha$23. It is worth noting that residues 514 to 517 are annotated as a $\beta$ sheet in the UniProt database (S1 Table), while these residues form an $\alpha$ helix in the reference structure used in this study (PDB ID: 6TLC). This result is not necessarily incongruous with the database

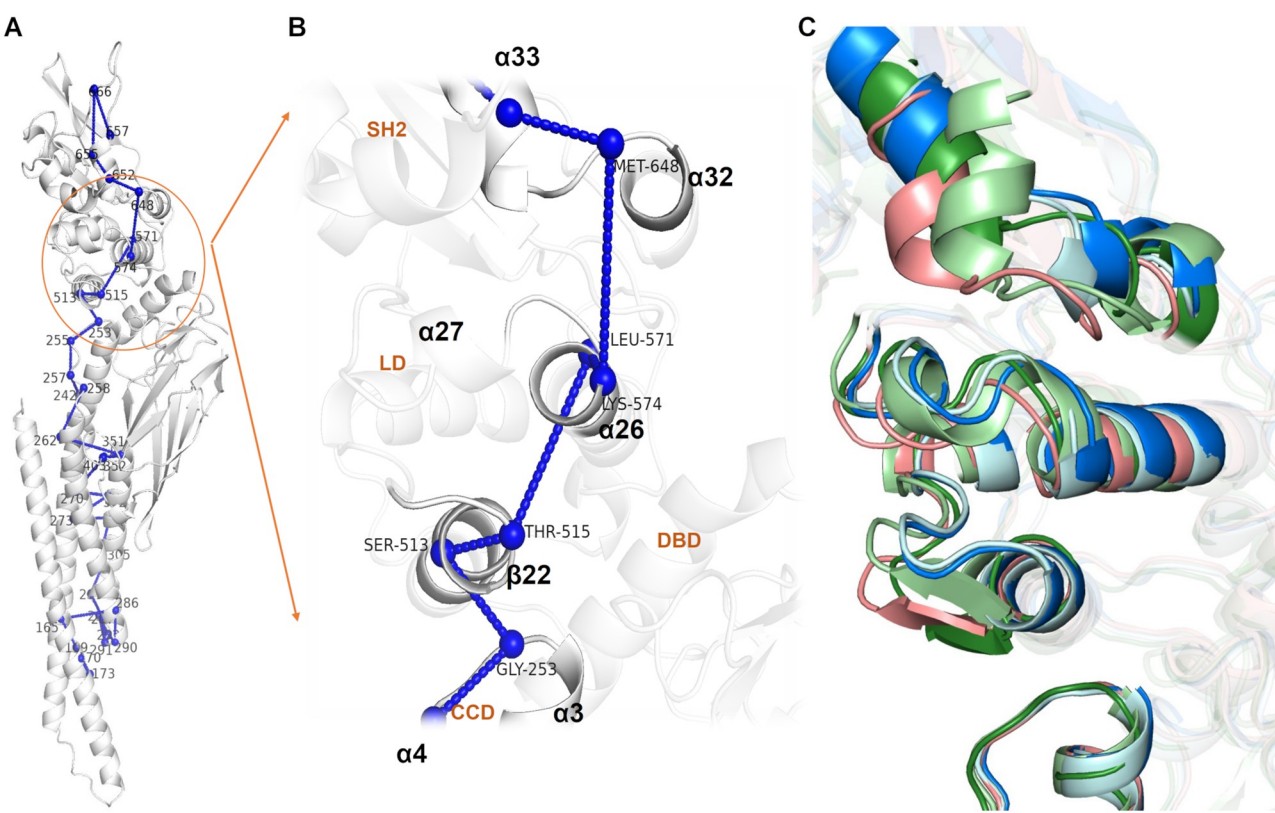

**Fig 4. Proposed allosteric path from D/A170 to Y657 obtained from REDAN analysis and its structural details. (A)** Residues involved in the allosteric path from CCD to LD and to SH2 domain. The raw path can be found in (S7 Fig). **(B)** A close-up view of signal transduction from CCD domain to SH2 domain. The residues identified by REDAN show a path through $\alpha$3 in CCD to $\beta$22 and $\alpha$26 in the LD domain and finally to $\alpha$32 and $\alpha$33 in the SH2 domain, bypassing DBD. **(C)** Average structures from all the macro-states show significant reorganization in this interface (colors as in Fig 2H). $\beta$ sheet configuration (adjusted in PyMol) of macro-states 2, 3, and 4 are shown to highlight rearrangement of $\beta$22 based on Ramachandran Dihedral for $\beta$-sheets of residues 514 to 517.

designation as multiple configurations of the secondary structure are observed experimentally. Stabilization of $\beta$22 as an extension of the $\alpha$21 helix in macro-states 0 and 1 stabilizes extension and a shift of the $\alpha$26 helix. The REDAN analysis suggests that further interaction between $\alpha$26 and $\alpha$32 then positions the $\alpha$32-$\alpha$33 loop such that $\alpha$33 adopts an extended conformation without close contact to the $\alpha$32-$\alpha$33 loop. On the other hand, the structural conformation analysis above suggests that $\alpha$26 may more indirectly affect $\alpha$32 via hydrogen bonding and hydrophobic interaction with $\beta$29.

**Hydrogen bond network.** The formation and breaking of hydrogen bonds plays an important role in stability of secondary structures and conformational variability of tertiary structures of a protein. To complement structural insights and REDAN analysis, the differential rate (preponderance) of hydrogen bond occurrence between different macro-states was compared (Fig 5). Hydrogen bonds were identified using Baker-Hubbard hydrogen bonding analysis, and then the differential hydrogen bond rate was calculated between macro-states 0, 1, 2, and 3, and the wild type-dominant macro-state 4. Macro-state 4 was considered as the primary active state as the wild type and D170A share high occupancy rates at this state. Note that Baker-Hubbard hydrogen bonding analysis algorithm classifies salt-bridge formation between amine and carboxylic acid as a hydrogen bond as well, thus the formation and breaking of salt bridges were also considered in the analysis.

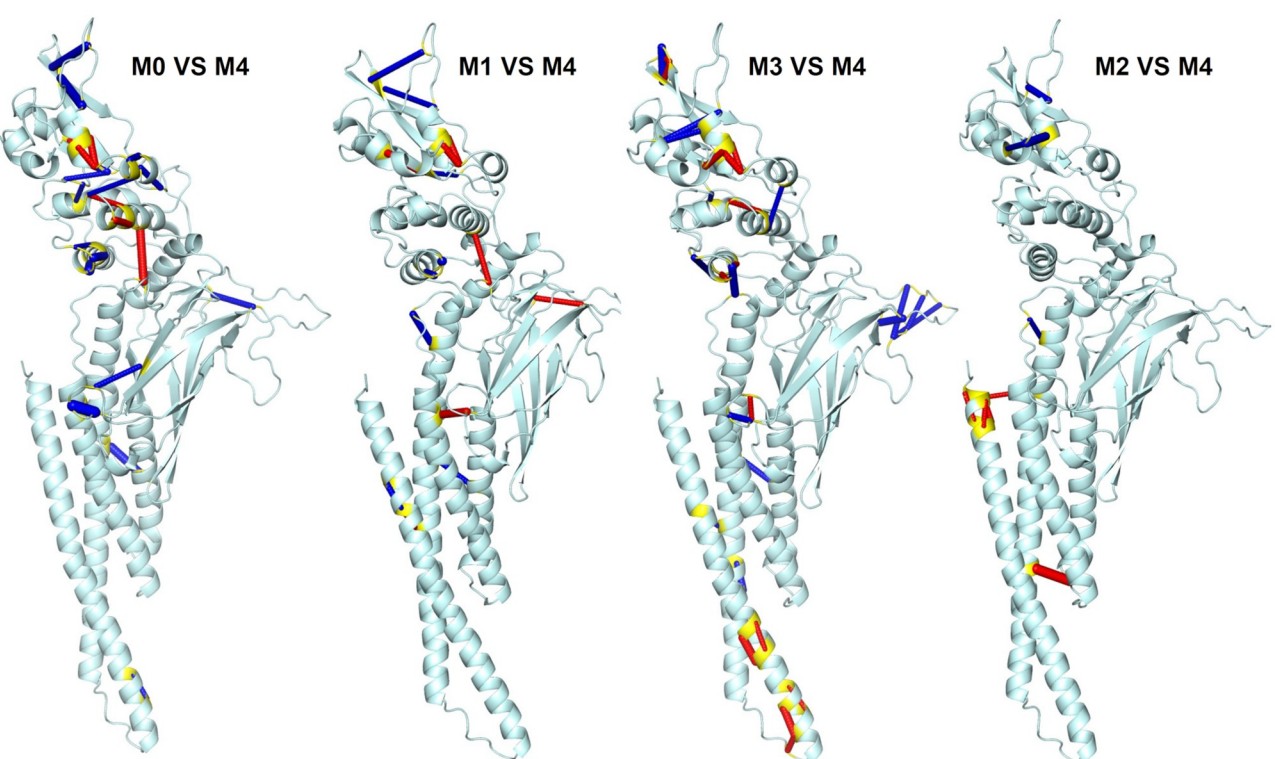

**Fig 5. Hydrogen bond frequency analysis showing key differences between macro-states.** The difference in the frequency of pair residue hydrogen bonds for macro-states M0–M3 compared to macro-state M4. Blue connections indicate that a hydrogen bond is present in at least 50% higher rate, while red connections indicate at least 50% lower rate with respect to macro-state M4. Terminal residues are colored in yellow. While significant changes in hydrogen bonding are evident across the protein, macro-states M0, 1, and 3 clearly indicate large changes in the pY+3 pocket, as well as consistent hydrogen-bonding changes linking the SH2 and LD domains.

Consistent with the observation from REDAN analysis, $\beta$22 has large changes in the hydrogen bond between the native (macro-state 4) and allosteric (macro-states 0 and 1) configurations. In macro-states 0 and 1, hydrogen bonds formation in $\beta$22 stabilizes the alpha helix, while the other macro-states alternate between an $\alpha$ helix and a $\beta$ sheet. Moreover, in macro-state 3, rearrangement causes new a hydrogen bond to form between $\beta$22 and $\alpha$3. These changes highlight key differences between the wild type and D170A variant as only the D170A variant occupies macro-states 0, 1 and 3 (Fig 5). These macro-states also show decoupling of LD to DBD along with alteration of interactions between LD and SH2 (Fig 6). First, a salt bridge between D566 (LD) and R335 (DBD) is lost in macro-states 0 and 1 (Fig 6A and 6B). The loss of this interaction allows for the shift of $\alpha$26 observed in the macro-states (Fig 4C). Second, a new hydrogen bond is formed between I576 (LD,$\alpha$26–$\alpha$27 loop) and N646 (SH2, $\alpha$32) in macro-state 0 (Fig 6A) and a salt bridge is formed between D570 (LD, $\alpha$26) and K642 (SH2, $\alpha$32) in macro-state 3 (Fig 6C). Third, loss of hydrogen bonds of E652 and/or I653 (LD, $\alpha$33) with S649 (LD, $\alpha$32–$\alpha$33 loop) in macro-states 0, 1, and 3 (Fig 6A–6C) contributes to increased flexibility of the $\alpha$33 helix, also as observed in Fig 4C.

Focusing on the upper part of CCD (Fig 6D–6F), it is clear that $\alpha$1 interacts directly with $\alpha$2 and $\alpha$4 through a large interaction surface. $\alpha$2 and $\alpha$3 form a contiguous helix interrupted by a kink in the helix at residue 278, thus perturbation of $\alpha$2 is structurally coupled with $\alpha$3. A salt bridge between E229 ($\alpha$2) and R306 ($\alpha$5) has a high appearance rate in macro-states 0, 1, and 3

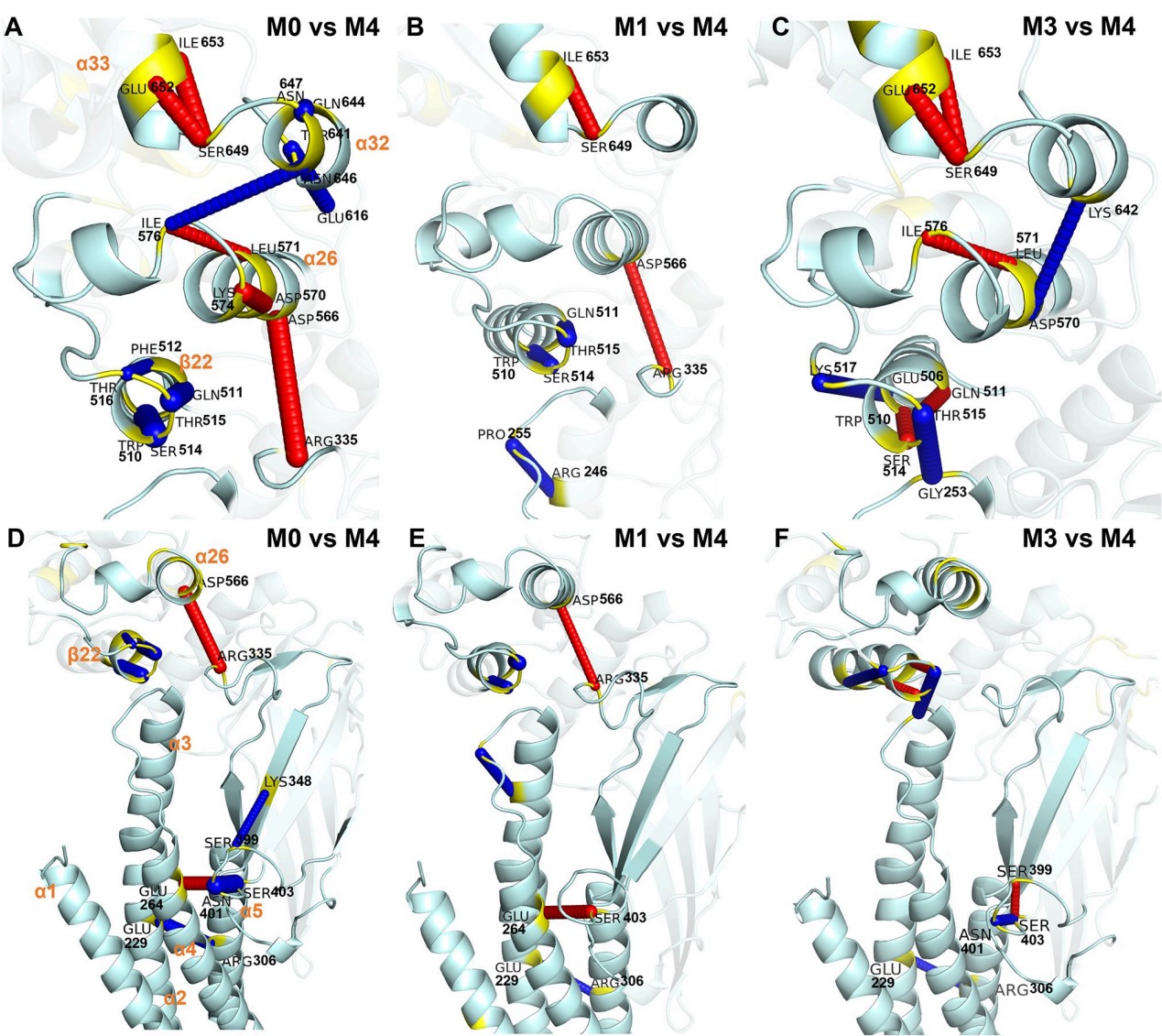

**Fig 6. A close-up view of the hydrogen bond network changes between macro-states M0–M3 and macro-state M4 (see Fig 5 for details). (A—C)** Differential rate of pair residue hydrogen bonds in the LD and SH2 domains. **(D—F)** Differential rate of pair residue hydrogen bonds in the CCD and LD domains.

compared to macro-state 4. This salt bridge provides a means to strongly correlate the motion of α2 to α5 and mitigate perturbations by α1. These interactions perturb α3 which are translated to the rigid core interface. The residues of α3 domain can thus be used to influence the rigid core to elicit interdomain response(S4 and S5 Figs). Moreover, I252 resides at the terminal end of α3 which forms strong hydrogen bond with Q511 (LD) identified in the rigid core analysis (Fig 3C). Q511 also forms a strong hydrogen bond with W474 (DBD). This network of hydrogen bonds allows a strong correlated motion between α3 in CCD and α20/α21 in the LD domain. I252 is also adjacent to E253 which was identified by REDAN analysis as part of the allosteric pathway thus providing a chemical basis for signal transduction from CCD to SH2 via the LD domain.

## Discussion

Significant differences in conformational space of the SH2 domain binding pocket (pY+3) between the wild type and D170A mutant were observed. The D170A variant explores and extends conformational space of SH2 domain, specifically with significant changes in opening of the pY+3 pocket. The six independent trajectories obtained for each variant explore fairly distinct areas of conformational space (i.e. different macro-states as identified in the clustering analysis, see Fig 2H). The long transition timescale between these states validates our kinetic clustering model, but also necessitates a "global" view of the trajectories in order to explain the full conformational dynamics of the protein. Based on the coverage of the 2D PCA space of the SH2 domain (Fig 2), we find that the six trajectories obtained are sufficient to sample the dynamics relevant to ligand binding affinity. On the other hand, the long transition timescale necessarily leads to high variability between trajectories for certain computed properties such as RMSF (S8 Fig). While not an indication of insufficiency in the global conformational view obtained from combining all trajectories, this variability instead serves as a valuable additional measure of conformational differences between kinetic macro-states. For transparency, the plots of replicates analysis are available in supplement(S2 and S6 Figs). Since the present MD simulation cannot guarantee sampling of the full conformational space of the protein, other possible allosteric pathways cannot be excluded. The mechanism proposed here is drawn with caution and the results agree with experimental observations. Thus, this work presents a theoretical foundation upon which to draw inspiration for drug design as well as mechanistic studies on the protein.

The hydrophobic environment formed by I659, W623 and F621 in pY+3 were previously shown to assist in binding of target peptide. Changes in the hydrophobic environment by increasing either hydrophobicity or aromaticity leads to hyper-activation, while introduction of polarity and reduction of hydrophobicity and aromaticity diminishes STAT3 function [27]. Furthermore, studies suggest that the side-chain Y657 interaction is important for stabilizing the ligand-protein complex [15–17]. Thus, the diminished binding affinity can be attributed to increased motions in the structures surrounding the pY+3 pocket in D170A. While in principle, translation of motion through the rigid core could affect the conformation of the primary pY binding pocket, only the pY+3 pocket shows differential motions in D170A compared to the wild type. Thus, our simulations support the conclusion that mutation of the D170 residue affects an inactivation of the protein via an allosteric mechanism resulting in conformational changes primarily in the specificity determining pY+3 pocket.

The observed differences in the SH2 conformational space allow us to further characterize the mechanism of this allosteric effect. The $\alpha 3$ helix of the CCD domain correlates functionally with SH2 conformations, as evidenced by strong correlated motions within the rigid core. Given its positioning, it likely communicates changes in the CCD to the LD. Using rigid body analysis, a potential pathway from CCD through LD, and finally to SH2 was identified, wherein conserved hydrogen bond networks and other strong interactions firmly link a series of secondary structures (primarily $\alpha 3$, $\alpha 20$, $\alpha 21$, and $\alpha 24$). While further analysis supports this proposed mechanism (vide infra), controlled mutagenesis of these key residues could also provide experimental evidence for the importance of these interactions to the D170A allosteric pathway. The lack of a significant increase (or decrease) in dynamic motion and overall flexibility in D170A compared to the wild type (as seen in the RMSF analysis) also supports a sequence of interactions between rigid bodies as the main allosteric mechanism.

The specific allosteric pathway was further elucidated via REDAN and differential hydrogen bonding analysis. These analyses both point to a very specific mechanism (structural details shown in S9 Fig): 1) stronger interaction between $\alpha 5$ and $\alpha 2$ causes a tilt in the $\alpha 2 / \alpha 3$ helix, 2)

$\alpha3$ tilt interfere the interaction between $\beta22/\alpha23$ and $\alpha26/\alpha27$. In macro-state 0 and 1, the interference breaks the salt bridge between D566 and R335 and caused $\alpha26$ shift away $\beta22$. While in macro-state 3, $\alpha26$ shift toward $\beta22$, which cause a steric clash between Lys 573 and $\beta22$. 3) in turn, the movement of $\alpha26$ and $\alpha26/\alpha27$ causes breakage of the hydrogen bonds between $\alpha33$ and the $\alpha32$-$\alpha33$ loop, 4) $\alpha33$ extends significantly and alters the conformation of the pY+3 pocket. We also identified conserved interactions between $\alpha26$ and $\beta29$ in SH2 which may provide further coupling.

This study independently confirms that alterations of the alpha-helical rigid core impacts the SH2 domain in the absence of crystal packing interactions. This supporting prior crystallographic observations indicating that small molecules, mutations, or monobodies can drive long-range allosteric changes to the SH2 domain through subtle alteration of rigid core. We have specifically avoided an elucidation of the allosteric pathway within the CCD domain (i.e. in the immediate vicinity of the mutation site). Although fully exploring comparisons to the allosteric pathways of other effectors (e.g. MS3–6, K116, MM-206) is beyond the scope of the present work, it seems highly likely that the observed structure of the interdomain interactions and the rigid core mechanisms should result in highly similar allosteric mechanisms within the LD, DBD, and SH2 domains. Conversely, the diverse nature of these effectors likely rules out any significant commonality in the initial few steps of the allosteric pathway. Additionally, we have observed that even fairly significant alterations of the CCD structure, such as the kinked conformation of $\alpha1$ explored by the D170A variant, do not correlate with changes to SH2 structure except via the rigid core. Thus, we consider this feature of the CCD structure as the "trigger" for the overall allosteric pathway.

Previously, the role of $\alpha26$ in allosteric communication has been identified: nuclear magnetic resonance (NMR) studies showed that mutation of I568F was able to induce a chemical shift perturbation in SH2, DBD, and CCD [28]. Besides, D566A, D570A and D570K mutants showed profound negative effects on transcription, and also unexpectedly tyrosine phosphorylation even before interleukin (IL) 6 induction [29, 30]. Additionally, previous study shows similar allostery pathways in STAT5, however rigid core was not explored [31]. STAT family of proteins are highly similar in primary, secondary and tertiary structures and the similarity in allosteric pathways of STAT3 and STAT5 can be used to posit that STAT family of protein have a rigid core that couples allostery between their domains.

These results present us with novel ways of regulating the CCD domain whereby ligand or peptide interaction with CCD can significantly alter the helical tilt of $\alpha3$ which is transmitted to the SH2 domain via the identified allosteric pathway. It is unlikely that different effectors will have identical mechanisms within CCD, as the local perturbation caused by point mutation, small molecule binding, peptide binding, etc. is radically different. However, our analysis supports the conclusion that any perturbation resulting in a change of $\alpha3$ tilt should result in a similar outcome due to the strong and highly concerted motions along the proposed allosteric pathway. Potentially, alteration of helical tilt could also result in tighter binding of target peptides to the SH2 domain while, as observed here, helix-helix interactions in CCD can also promote structural changes in SH2 domain leading to reduced affinity of SH2 binding. In short, the mechanism for signal transduction has been identified through analysis of dynamic correlated motions between CCD and SH2 domain, the exact outcome of this allosteric effect is not clear. Distinguishing between two potential mechanisms: changes in substrate specificity to form heterodimer required for phosphorylation of Y705 and reduction in homodimerization due to reduced affinity towards pSTAT3, needs to be investigated. Drugs designed to specifically alter the motions of the CCD helices would yield valuable insight towards validation of proposed mechanism. Since the pY+3 pocket regulates the affinity of peptide binding, rather than being the catalytic active site, assays developed to probe allosteric regulation of SH2 via

CCD should consider the identity of peptides used to target the SH2 domain in addition to overall activity.

Finally, the methods used here to identify the rigid core mechanism, specifically a combination of structural and conformation analysis (dimensionality reduction, functional clustering, and conserved $C\alpha$ pair distances) with dynamical and correlative analyses (REDAN and differential hydrogen bond analysis) should allow for identification of other potential effectors targeting CCD, and more widely, in identifying similar allosteric pathways in a number of (semi-)rigid proteins.

## Methods

### Initial structure

The monomer STAT3 with peptide MS3–6 complex (PDB ID: 6TLC) structure was used as the template for this study. In chain A, residue 372–381 and 418–428 in DBD domain were missing, here these residues were modeled using Chimera [32]. The apo structure was created by directly deleting the MS3–6 peptide, and then the apo type was subjected to mutation using the PyMol Mutagenesis Wizard [33] to generate the D170A variant. To show DNA binding (Fig 1), the 1BG1 structure was used which contains a single STAT3 bound to DNA. However, STAT3 binds to DNA in its dimeric state; the dimeric structure was generated in PyMol to show DNA binding of STAT3 dimer. Hydrogen atoms were added to the crystal structures using PyMol. The protonation states for the histidine residues were assigned using the H++ program [34].

### Molecular dynamics simulation

For each system, a rectangular periodic water box with 84736 TIP3P waters was used with a minimum distance of 10 $\mathring{A}$ between the box boundary and the protein to avoid image interactions(S12 Fig). To balance charge and provide realistic salinity, 0.15M sodium and chloride ions were added. NAMD 2.13 [35] with the CHARMM 36 force field [36] was used for energy minimization and molecular dynamics (MD) simulations. Initially, the simulation systems were subjected to 5000 steps of energy minimization to remove bad contacts and clashes. Then, systems were heated from 0 K to 300 K, heating 50 K every 200 ps, and then from 300 K to 310 K in 200 ps, with 10 ns isothermal-isobaric ensemble (NPT) short equilibration. Subsequently, six replicas of 600 ns canonical ensemble (NVT) MD simulations at 310 K were conducted. The first 100 ns simulations were discarded as equilibration and the following 500 ns for each replica, 3$\mu$s in total, was used for further analysis. The SHAKE algorithm was applied to all bonds containing hydrogen atoms. The electrostatic interaction was evaluated by the particle-mesh Ewald method, and Lennard-Jones interactions were evaluated using 10 $\mathring{A}$ as a cutoff. The NPT simulations were performed using a Nosé-Hoover Langevin piston pressure. The NVT simulations were performed using the Langevin integrator. For the integrator, a friction coefficient of 1 ps-1 was implemented. A step size of 2 fs was used.

NVT was used for the main simulation due to its propensity for greater stability over longer timescales. To reconcile the change in system from NPT to NVT, total energy, kinetic energy, potential energy, temperature and pressure of the system were examined for consistency and smoothness. Both systems were well equilibrated (S10 Fig). Sufficient sampling of the MD simulation for both wild type and D170A variants was evaluated by the pair Root Mean Square Root (RMSD) (S11 Fig), which shows that the simulations have either reached a stationary shape or there is transition between different stationary shapes. Detailed RMSD and RMSF analysis can be found in the Supplemental Information (S1 Text).

### Feature characterization

Various feature characterization relevant to trajectory analysis were conducted using two open-source packages: MDTraj 1.9.3.40 [37] and MDAnalysis [38, 39].

**Pair residues center of mass distance.**   Pair residues center of mass distance was used to characterize the SH2 domain PY pocket and PY+3 pocket. First SH2 domain was extracted using atom_slice function from MDTraj; then the residue center of mass was calculated using the `center_of_mass` function from MDAnalysis, and finally the Euclidean distance of pair residues center of mass was calculated.

**Inter-domain pair C$\alpha$ distances.**   The inter-domain pair C$\alpha$ distances were used to carry out the rigid core analysis. Take CCD-DBD domain pair C$\alpha$ distances for example, first, the neighbors C$\alpha$ atoms from DBD that are within 1 nm of CCD were found using compute_neighbors function from MDTraj, from which we can find the pair residues between CCD and DBD that are within interaction ranges. then the common pair residues between wild type and D170A variant were kept, and the pair ca distance were calculated using `compute_distances` from MDTraj.

**Alpha helix global tilt.**   The $\alpha$ helix global tilt angle was used to characterize the geometry of helices according to the procedure of Sugeta and Miyazawa [40]. After alignment the trajectories to the crystal structure, the `helanal.helanal_trajectory` function from MDAnalysis software was used to characterize the alpha helices. [0, 0, 1] was used as the reference axis [41].

### RMSD analysis

Root-mean-square deviation (RMSD) analysis shows the conformational dynamics over the trajectory and provides an insight into the variation within the conformational space of a reference structure. After alignment to the reference structure, the RMSD values were measured using MDAnalysis.rms function.

### RMSF analysis

Root-mean-square fluctuation (RMSF) allows us to probe average positional changes of each residue. RMSF measures the average deviation of a particle (and individual residue) over time from a reference position (typically the time-averaged position of the particle). The trajectories were first superposed to the backbone of first frame of each trajectory, then the RMSF values for each trajectory were measured using `mdtraj.rmsf` using MDTraj. Finally, the mean RMSF values and the standard error of each residue over the six replicas were calculated and plotted.

### Principal components analysis

Linear Principal Components Analysis (PCA [42]) is used to transform high dimensional and often linearly-dependent data points into a low-dimensional space spanned by uncorrelated principle components. The first two principal components (PCs) were used in this work, yielding a two-dimensional reduction of the original data. Given high dimensional data represented in n (sample size) by m (variable size) matrix, the covariance of any two variables X and Y was calculated by,

$$\mathrm{cov}(X_{i\alpha}, X_{j\beta}) = \frac{N}{N-1} \langle (X_{i\alpha} - \langle X_{i\alpha} \rangle) \cdot (X_{j\beta} - \langle X_{j\beta} \rangle) \rangle,$$

where $X_{i\alpha}$ is the $\alpha$ Cartesian component of the coordinate vector for atom $i$. The covariance

matrix $C$ is constructed as the pairwise covariance between all variables. The eigenvectors of $C$ are the components of PCA, while the eigenvalues measure the contribution of each PC in the dataset. The eigenvectors also provide a mapping from the high-dimensional dataset to the low-dimensional PC space: the (PC1,PC2) coordinates for each input frame are given by multiplication of the original data set by the first two important eigenvectors. For a given PC, the importance of a feature (variable) is reflected by the absolute magnitude of the corresponding entry in the eigenvector. The PCA analysis was performed by Scikit-learn [43] implemented in Python.

## Markov state modeling and Perron cluster cluster analysis

Markov state models (MSMs [44, 45]) have shown great utility in modeling the transitions among functional states. It was used in this study to cluster conformations into kinetically meaningful macro-states. The conformational space was first discretized into $n$ micro-states. Here, agglomerative clustering [46] was applied to divide the sampled conformations into 300 micro-states in the two-dimensional PCA coordinate system. Then, $C_{ij}(\tau)$, the number of observed transitions from micro-state $i$ to micro-state $j$ at a lag time $\tau$ is calculated. Then the transition probability, $P_{ij}$, from micro-state $i$ to micro-state $j$ can be estimated as $P_{ij} \approx (C_{ij} + C_{ji})/\Sigma_k(C_{ik} + C_{ki})$. According to estimated relaxation timescale (S1(B) Fig), the count matrix and MSM transition probabilities converge beyond $\tau = 5$ ns, which was chosen as the lag time.

Finally, Perron Cluster Cluster analysis (PCCA), implemented in the PyEmma package [47], was be used to coarse grain micro-states to "kinetically relevant" macro-states based on the well-sampled micro-state transition matrix. Structures that interconvert frequently were assumed to belong to the same functional metastable state (macro-state)[48]. Five macro-states were determined based on the band gap in the estimated relaxation timescale plot.

## Relative entropy-based dynamical allosteric network

The relative entropy-based dynamical allosteric network (REDAN) model [49] was used to quantitatively characterize protein allosteric effects upon mutation. The difference between the distributions of the pair alpha carbon (C$\alpha$) distances of two residues upon perturbation is quantified by the perturbation relative entropy (PRE), which is the average relative entropy,

$$\text{PRE}(P||Q) \quad = \frac{D_{KL}(P||Q) + D_{KL}(Q||P)}{2},$$

$$D_{KL}(P||Q) \quad = \int_0^\infty p(x) \ln \frac{p(x)}{q(x)} dx,$$

where $p(x)$ is the distribution density for system $P$ (before perturbation), and $q(x)$ is the distribution density for system $Q$ (after perturbation). High perturbation relative entropy values indicate that substantial allosteric effects are implied by the significantly different distance distribution of the residue pair.

Then, a weighted graph can be built based on the PRE matrix. Each node is represented by a C$\alpha$ atom, and two nodes will be connected by an edge if the longest possible distance between them is less than 10 $\mathring{A}$. Each edge is weighted as 1/PRE. Since high PRE values indicate importance in the propagation of the structural changes in the protein, the pathway with the smallest overall weight implies the most structurally relevant route and hence a possible allosteric pathway. Dijkstra's algorithm was used to identify the shortest pathway.

## Supporting information

**S1 Text. RMSD and RMSF analysis.** D170A mutation induces large structural but minor dynamical changes in STAT3.**Table A.** Statistical differences for the $\alpha$3 global tilt among different macrostates ([Fig 3F]). Kolmogorov-Smirnov test and T-test were done using scipy.stats.ks_2samp and scipy.stats.ttest_ind function respectively.
(DOCX)

**S1 Table. Secondary structure assigned by the UniProt database by compiling structure information from multiple x-ray crystal structures.**
(TIF)

**S1 Fig. Additional information of PCA and MSM analysis. (A)** PCA scree plot: dot shows the cumulative explained variance of the principal components; the bar chart represents the explained values per component. **(B)** Relaxation timescales of MSM for SH2 domain conformational space at different lag times. **(C)** The first ten features that contribute the first two PC the most. The absolute value of PC1 and PC2, and square root of sum of squared PC1 and PC2 values are shown here.
(TIF)

**S2 Fig. Pair C$\alpha$ distance distribution between Q644 and Y657, Q644 and E638 of 6 replicas for wild type and D170A variant.**
(TIF)

**S3 Fig. Characterization of CCD using pair C$\alpha$ distances. (A)** PCA 2D plane of CCD pair C$\alpha$ distances colored by macro-state from SH2 domain results. **(B)** Represent structure of CCD corresponding to Figure A. **(C)** The coefficients of first 100 features that contribute the first two PC the most.
(TIF)

**S4 Fig. The PCA analysis of pair distances of rigid core (residues:240–252, 474, 479, 511, 546, 549, 550, 562,564, 568, 610, 611). (A)** PCA 2D plot colored by different macrostates; **(B)** PCA 2D plot colored by systems; **(C)** The first ten features that contribute the first two PC the most. The absolute value of PC1 and PC2, and square root of sum of squared PC1 and PC2 values are shown here.
(TIF)

**S5 Fig. The PCA analysis of pair distances of rigid core without residue 562 (residues:240–252, 474, 479, 511, 546, 549, 550, 564, 568, 610, 611). (A)** PCA 2D plot colored by different macrostates; **(B)** PCA 2D plot colored by systems; **(C)** The first ten features that contribute the first two PC the most. The absolute value of PC1 and PC2, and square root of sum of squared PC1 and PC2 values are shown here.
(TIF)

**S6 Fig. CCD $\alpha$3 global tilt angle distribution. (A,B)** CCD $\alpha$3 global tilt angle distribution of different macrostates, plotted separately for the wild type and D170 variant. Average helix tilt angle within each macro-state is illustrated by a vertical dashed line. **(C,D)** CCD $\alpha$3 global tilt angle distribution of different replicas for wild type and D170A variant.
(TIF)

**S7 Fig. Additional REDAN analysis results. (A)** Proposed pathway from 170 to 640 shown in the protein structure; **(B)** Proposed pathway from 170 to 644; **(C,D,E)** Key pair residue

distance; **(F)** Ramachandran Dihedral for residue 513,514, 515 and 517; **(G)** Summary of proposed pathways from source residue 170 to target residue 640,644 and 657.
(TIF)

**S8 Fig. The mean RMSF of six replicates was plotted, with the RMSF values for each residue separated by domain.** The wild type is plotted in blue and the D170A variant in orange, with error bar indicating the stand error among the six replicates. Structures of each domain colored by wild type RMSF values are shown (low RMSF values in white to high RMSF values in red).
(TIF)

**S9 Fig. Proposed specific mechanism. (A)** Representative structure macro-state 0 (light cyan) compare with macro-state 4 (salmon); **(B)** Representative structure macro-state 3 (green) compare with macro-state 4 (salmon) **(C-G)** key residue pair CA distance or contact distance (closest heavy atom distance). ILE-252 to LYS-573 distance distribution and PHE-512 to LYS-573 distance distribution show $\alpha$26 shift away $\beta$22 in macro-state 0 and 1, while toward in macro-state 3; PHE-512 to LEU577 and PHE-512 to SER-649 distance distribution show $\alpha$26/$\alpha$27 and $\alpha$32/$\alpha$33 loops move away $\beta$22 in macro-state 0, 1 and 3, while in macro-state 3, $\alpha$32 moves close to $\alpha$26 as indicated by ASP-570 to LYS-642 distance distribution.
(TIF)

**S10 Fig. System parameters (pressure, temperature) and system energy (potential energy, kinetic energy and total energy) of wild system and D170A variant system, each system has six replicas: cp1, cp2, cp3, cp4, cp5 and cp6.**
(TIF)

**S11 Fig.** First row: Root Mean Square Deviation (RMSD) of wild system and D170A variant system, each system has six replicas: cp1, cp2, cp3, cp4, cp5 and cp6. Note: rolling average of every 100 frames was plotted here for better visualization. Second row: The pair RMSD (frames every 1 ns were extracted and used for pair RMSD calculation) of both systems for each replica.
(TIF)

**S12 Fig. Figure depicting the simulation system, generated by VMD.** Water is shown as lines, ions are shown as vdw, protein is shown as cartoon.
(TIF)

**S13 Fig. Root Mean Square Deviation (RMSD) analysis. (A, B)** Cross-correlation (Pearson correlation) of RMSD values of each domain for wild type and D170A variant. Here RMSD values were calculated with the first frame of reference since the correlation of dynamic changes of each domain is of interested. The cross-correlation was done by merging the rmsd of 6 copies trajectory together, and then the Pearson correlation among different domains were calculated. While worth being noted, the correlation value does not suggest the functional correlation between domains, since RMSD is an overall measurement of conformational changes with regarding to the reference structure, distinctive conformations may have same RMSD value. **(C)** Violin plot of RMSD values for the whole protein (core full length protein, not including NTD) and each domain. Here crystal structure was using as reference since the conformational changes difference between wild type and D170A was of interested. **(D)** The RMSD distribution of SH2 domain in the six replicas for D170A variant.
(TIF)

## Acknowledgments

TZ was supported by a fellowship from the Department of Chemistry of Southern Methodist University. All calculations were performed on the ManeFrame II supercomputing system at Southern Methodist University.

## Author Contributions

**Conceptualization:** Tingting Zhao, Nischal Karki, Brian D. Zoltowski, Devin A. Matthews.

**Data curation:** Tingting Zhao.

**Formal analysis:** Tingting Zhao.

**Funding acquisition:** Devin A. Matthews.

**Investigation:** Tingting Zhao.

**Methodology:** Tingting Zhao, Nischal Karki, Brian D. Zoltowski, Devin A. Matthews.

**Project administration:** Devin A. Matthews.

**Supervision:** Brian D. Zoltowski, Devin A. Matthews.

**Visualization:** Tingting Zhao.

**Writing – original draft:** Tingting Zhao.

**Writing – review & editing:** Tingting Zhao, Nischal Karki, Brian D. Zoltowski, Devin A. Matthews.

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
