## [Decision Letter · Decision Letter 0]

13 Jul 2022

Dear Ms. Zhao,

Thank you very much for submitting your manuscript "Allosteric Regulation in STAT3 Interdomains is Mediated by a Rigid Core: SH2 Domain Regulation by CCD in D170A Variant" for consideration at PLOS Computational Biology.

As with all papers reviewed by the journal, your manuscript was reviewed by members of the editorial board and by several independent reviewers. In light of the reviews (below this email), we would like to invite the resubmission of a significantly-revised version that takes into account the reviewers' comments.

We cannot make any decision about publication until we have seen the revised manuscript and your response to the reviewers' comments. Your revised manuscript is also likely to be sent to reviewers for further evaluation.

Sincerely,

James Gallo

Associate Editor

PLOS Computational Biology

Nir Ben-Tal

Deputy Editor

PLOS Computational Biology

Reviewer's Responses to Questions

**Comments to the Authors:**

Reviewer #1: Overall:

While I do not feel qualified to comment on the importance of understanding STAT3, or the current knowledge regarding this protein, I am qualified to evaluate the molecular dynamics simulations and analysis. I found this manuscript to be engaging, and I believe that this study has high scientific rigor. The authors were very diligent and careful in their simulations and analyses.

My primary criticisms, as you will find detailed below, primarily deal with the clarity of the take home message. I think there are things that should be removed from the paper because they confound the story. Also, some effort should be put into clarifying figures and text.

Significant Suggestions:

While this is a significant modification, it will be an easy suggestion to implement. In the Methods section, there is no need to give equations or elaborate on details that are taken care of by the software. For example, for RMSD, just tell the reader what the RMSD measures in a sentence or two, and then tell them the software (MDtraj) and the command(s) you ran (presumably mdtraj.rmsd). If there are cases where you actually generated values and then plugged them into formulas, then tell us, but if it’s baked in the software then why bother.

An overall suggestion that encompasses several points below is to ask the following question for each figure/subsection: Does this support and clarify the take home message by adding a new point, or is it unnecessary? If it is unnecessary, then don’t include it in the manuscript. I think in some cases (e.g., perhaps RMSD, RMSF), these analyses led you to the more advanced (e.g., MSM). However, if they don’t add anything to the story, then it would be better to include them in a supplement than the main text. Otherwise, it confounds the main story and readers are left trying to understand what they are supposed to learn from a figure. A specific example of this is the RMSD analysis in the paper. I certainly do understand why you would have done this analysis as part of the study, but it doesn’t appear to add any value to the story that isn’t more clearly illustrated later. Especially given the point you make yourself in the caption, that different conformations can have the same RSMD. I would suggest removing this, or moving it to a supplement.

In fig 3 you chose to use the deepness of the color to represent standard deviation. I would suggest two modifications to this choice. (1) Use error bars rather than color depth. This allows the reader to easily assess the significance of the data points where there is a difference between the mutant and wt. (2) Show standard error rather than standard deviation (i.e., stddev/sqrt(6)). Again, this better allows us to evaluate the significance of the differences. The standard error quantifies the precision of your RMSF values rather than the spread of the RMSF values in individual runs. Also, some more minor points regarding fig 3. Please show the location of the mutation site. The whole point of the figure is comparing wt and mutant, yet I can’t tell where the mutation occurs from your figure. In the caption I assume “stand deviation” is meant to read “standard deviation”. Also, some font sizes are too small. Increase fonts sizes for axis labels, domain/site labels and the key in the first panel. Finally, I find the domain labeling to be confusing. What does \\alpha1-\\alpha2 mean? Why not just put the \\alpha1 label within the dashed lines that represent the boundaries for that sub-domain?

In fig 4 there are some hard to read fonts. In particular, the labels in panel D, G are hard to read. The yellow and orange pY and pY+3 labels could perhaps be outlined in black to enhance readability. All labels in these panels should use larger (and perhaps bold?) fonts. Also, it is not clear why you compared to 6nuq here when you were using a different PDB structure up to this point.

I found fig 5 to be very confusing. On page 13 you state “The existence of such an interaction network is demonstrated in Figure 5B, where the inter-domain pair Cα distances are plotted and colored according to standard deviation values computed across all trajectories.” I think you mean 5A here, but it still doesn’t make sense to me. For each plot you show 5 curves (M0-M4) and they appear to be color coded by this, not the standard deviation. M0-M4 is not defined anywhere. Reading ahead it looks like you might be referring to the macrostate number here? Anyway, the figure caption and manuscript text need significant clarification. A couple more minor points, I would suggest changing your language regarding “hydrogen bonds distance”. I think what you mean to say is distances between atoms that form hydrogen bonds. The reason for this change is that your distributions reach well past what most people would consider to still be a hydrogen bond, i.e., the bonds appear to be breaking sometimes. Finally, there are again some very small fonts. In panel B, there is a sub-plot that is quite literally unreadable at any zoom level. As before, consider both font sizes and colors.

The analyses presented in fig 5 and fig 7 seem to have a lot of overlap. While fig 5 includes some rigid core analysis, the hydrogen bonding part has a similar take home message (but is less clear to me) as fig 7. It is not clear why both are needed. I’m not saying you should eliminate fig 5, but perhaps consider removing the parts/panels that are redundant.

It is mentioned that MSM analysis was performed. Fig 4 shows some clustering results, but there is no mention of the transition probabilities in the main text. Does this analysis not add anything to the story?

Minor points:

*Page 2: “amino-terminal domain ()”, should there be something in the parentheses?

*Fig 1 caption: NTD not defined, perhaps this ties into the first minor point?

*Fig 1: Consider showing the location of the mutation site D170 that you are studying. I ended up figuring it out from going to the PDB because I wanted to know when looking at this figure.

*Page 6: “RMSD stabilized around 4”. Without looking at the figure I would assume angstroms, but you need units here.

*Fig 6: Residue labels in panel A are hard to read. Making them bold might help, but part of the issue is that some are very crowded and some are obscured by the dark blue line.

Reviewer #2: In the paper “Allosteric Regulation in STAT3 Interdomains is Mediated by a Rigid Core: SH2 Domain Regulation by CCD in D170A Variant”, Davin Matthews and co-workers analyze by means of Molecular Dynamics simulations the effect of the D170A changing on the dynamics of the STAT3 protein. In the paper, they put in evidence that the two variants of the protein populate a different conformational space. In particular, a different conformational state of the helices in the Coiled-Coil domain populated in the D170A variant affects the dynamics of the pY+3 binding site in the SH2 domain. Overall, the reported data seems to be convincing, however, in my opinion, in some cases, the statistical relevance of the reported data needs to be improved to prove that the discussed picture is not a consequence of stochastical fluctuations of the simulated systems. The authors acquire six different independent trajectories for each variant; inter-trajectories variability could be inserted to make the discussion more convincing. Another weakness of the paper that could be strengthened is the lacking of a discussion of a clear effect of the mutation on the interaction with its surrounding residues. One of the conclusions of the authors is that more than a different induced flexibility, the change in the residue 170 acts through a “rigid core mechanism”. This mechanism is described by proposing a clear sequence of interactions between the coiled-coil and the SH2 domains. The table in figure S4G clearly indicates this sequence of interactions, but, if this is the case, a clear different picture in the nearest-neighbor interactions of the residue 170 in the wild-type protein and in its D170A variant should be observed. Quite surprisingly, the authors do not search for these differences; I think that these data could contribute to making the hypothesis of the authors more convincing. Finally, my last major concern regards the linking to the experimentally observed effect of the D170A changing. What is experimentally observed is a lowering in the affinity of mutants with respect to the wild type. What the authors observe in their dynamics is higher flexibility in the pY+3 binding pocket in the D170A variant. The authors would to better discuss this aspect, for example by showing that in similar systems a correlation between these so different observables has been demonstrated.

Here below, I report minor concerns/comments:

- The paper needs an editing revision. For example: i) at page 3, in the first lines of the introduction section, the acronym for amino-terminal domain is lacking; ii) in the caption of table s1, uniport instead than uniprot is reported; iii) at page 12, figure 5H does not exist, probably the authors refer to figure 5F; iv) In the methods section (in the first paragraph) the reported pdb code is wrong (6tle instead than 6tlc); v) at page 24, “is calculated” is repeated two times

- In figure 1B, residues clearly folded in helical structures are colored in yellow, a color that indicates unstructured regions (loops) according to the caption. Is the structure wrong or the problem is in the assignment?

- At page 4, in the description of the SH2 domain family, the authors mention the pY+3 interactions a as specificity region. This is probably the case for the SH2 domain in STAT3, but, this statement is far to be general, and wider regions (from -2 to +5) can be determinants for the selectivity (see for example the case of the N-SH2 domain in the SHP2 phosphatase). More in general, the discussion at page 8 seems to develop on two different planes, that of the SH2 domain in STAT3 and that of the SH2 domain in general. The authors could be more clear, defining each time at which SH2 they refer.

- - At page 4. The STAT3 regulation mechanism mainly discussed in the paper is that the phosphorylation of Y705 promotes STAT3 dimerization, by binding of the pY705 in the SH2 domain of another STAT3 protein. The general motif for the SH2 domain in STAT3 is described as pTyr-Xaa-Yaa-Gln. Quite strangely, the sequence surrounding Y705 (SAAPpYLKTKF) does not contain this motif. This point resembles my previous comment on the effect of mutation on the binding affinity. A change in the pY+3 binding pocket could reflect changes in selectivity more than in affinity. Commenting on this aspect could help the discussion.

- a brief discussion on the biological interest (if any) for the D170A variant could be of interest (e.g. is this variant causative of pathologies?).

- The authors use the RMSD from a single structure to prove that the system is well-equilibrated. This approach could be misleading (see for example: J. Chem. Phys. 1998, 109:10115 or Comput Math Methods Med. 2012; 2012:173521). Flat (sometimes high) values of the RMSD reflect only that the sampled conformations are all equally distant from the reference but little can be said about the distance between them. RMSD computed by comparing all the frames to each other (with results reported as a graphical matrix) could be more reliable. In this context, quite surprisingly, the RMSD values seem to be higher for the SH2 domain in the wild-type simulations than in those of the mutant. How do the authors comment on this evidence?

- Some regions of the protein have been modeled because absent in the pdb file. In the scheme below the figure 1A (in which the different domains are indicated) the modeled regions could be somehow put in evidence. This information is practically absent in the paper, but it can help to interpret the RMSF data.

- At page 7, the statement " D170A variant has multiple quasi-stable conformations seen as multiple peaks in the core full length protein violin plot" in the comment to figure 2c needs further comments. The time evolution of RMSD (with respect to a different structure) reported in figure S1 seems to suggest that this behavior is not homogeneous on the six replicas. Probably the RMSD distribution in the six replicas could be helpful to understand if only a few simulations are responsible for these differences or if they are observed in all the simulations

- in a comment on the RMSD data the authors write " Overall, the deviation of the SH2 domain from the crystal structure is significantly increased and several quasi-stable conformations appear, while the CCD domain becomes somewhat more rigid." RMSD takes into account differences between structures, and much less about flexibility. Under this point of view, RMSF is definitely better,. By the way, the RMSF values seem to suggest the opposite behavior.

- data in figure 2 are acquired from six different simulations. How much is the standard deviation between the different simulations? These data should be added to figures 2A and 2B. For example, the reliability of the statement "This correlation is enhanced at the expense of correlation of different domains to the linker domain, with only CCD retaining its correlations with linker in D170A variant." depends on the values of the standard deviations.

- by looking at figure S2, the distance between 644 - 657 and 638-644 can be used as an approximate representation of PC1 and PC2 (as also shown in figures 4 B, C, E, and F). The plot of these two distances in the 12 simulations could help to understand how the PC1-PC2 space is populated in the different simulations

- the sentence " The motion of Y657 and Q644 yields the largest coefficients across both PC1 and PC2, underlining their conformational importance" at page 9 is misleading. The authors refer to the addition of the two contributions to PC1 and PC2 but the distance contributes only marginally to PC2

- How much of the explained variance is described by PC1 and PC2 vectors in the PCA analysis reported in figure S3?

- At page 12 "While the rigidity of the α helices diminished the utility of PCA analysis, it also demonstrates that the dynamics of the helices are highly correlated (Figure 3)." How does figure 3 demonstrate the correlations between the dynamics of the helices?

- It is not clear if the data reported in figure 5 refer to WT, to the D17A variant, or both of them.

- figure 5B is hard to read. A table reporting the percentage of existence for all the considered interactions would be more useful

- I'm not sure that the data reported in figure 5F are statistically significant. What about these data in the single variants? Are these data similar in the six replicas for each system?

- Are REDAN analyses performed on both the proteins (wild type and D170A variant)?

- the color code used in figure S4F is not clear

- in the discussion, the authors write that " The D170A variant explores and extends conformational space of SH2 domain, specifically with significant changes in opening of the pY+3 pocket.". SASA data could be useful to strengthen this statement.

- in the methods section or in the figure 1 caption an explanation of the method used to obtain the structure in figure 1 should be inserted

Reviewer #3: In this manuscript, authors investigated the structural and dynamic features of wild type and mutated STAT3 protein using molecular dynamics simulations. The comparison between the wild and mutated simulations provided not only the structural and dynamical difference but also the information on allosteric mechanism. This work is interesting to the readers of PLOS computational biology, it would strengthen the paper if the authors could address the following points.

Major points:

1) In this paper, the result and discussion parts are separated. The results include inferences based on the other research; I think it is okay to include such descriptions. However, some of the descriptions seems to hinder readability. For example, P13 line 4-5 “the rigidity transmission was oberserved in other proteins (Sljoka 23 , Ye et al. 24 ). Thus we further hypothesized…”. I found some of the explanations in the result part difficult to follow; as I suspect a reader less familiar with this protein might have even greater difficulties, authors should make the manuscript and the figures easier to understand for readers.

2) The authors should correct the title of result section more appropriately.

3) P6-, the authors also analyzed the whole structural change. I suggest that a domain analysis (for example, dyndom, etc) using those representative structures may support your results.

4) P9, line 4-7, the sentence “From these qualitative observations, it can be inferred that the increased flexibility in CCD due to loss of a negatively charged residue D170, leads to an allosteric increase in flexibility of the DBD and a decrease in flexibility of the LD and SH2 domains.” is difficult to understand. They may add a more detailed explanation based on the results of RMSD and RMSF.

5) P11 line 2 and Figure S2C, I understand that “The ten largest coefficients of the first two PCs” means the eigenvectors of PCA. Is this okay? If so, is it correct that these values of Figure S2C positive values? (absolute values?)

6) P11 line 18, in this sentence “Besides, Zhang et. al. observed that the truncation of STAT3 to exclude α1 helix is fully capable of DNA binding upon tyrosine phosphorylation by Src kinase in vitro, suggesting the preservation of a functional conformation of the pY pocket. This agrees with our observation of a stable pY pocket in the D170A variant”, can the author explain the relationship between pY+3 pocket structure in simulations and this experimental data?

7) P12 last part -P13, I think that the sentence “Significant differences in global helical tilt for each of the macro-states of the SH2 domain is observed in α3 showing functional correlative differences among macro-states (Figure 5H). The α3 helix also interfaces with most of the other domains through its C-terminal helical turn. Residues at this interface could transmit motion through interacting residues from other domains.” is very important point. Does this data (Figure 5H->F?) show statistically reliable differences? Authors should add a more detailed explanation on this.

8) P13 line 7-9, in the sentence “The existence of such an interaction network is demonstrated in Figure 5B, where the inter-domain pair Cα distances are plotted and colored according to standard deviation values computed across all trajectories”, I suggest that authors may add a more detailed description on analysis of inter-domain pair Ca distances.

9) In methods (P23~), the authors should add a more detailed description (number of water molecules, etc) and figure on the simulation system. In addition, they should also add the figure to explain locations of the D170A, pY pocket, and pY+3 pocket

10) P23, why did the authors perform NVT simulations as production runs.

11) P24, the authors should add a more detailed description of cross-correlation and RMSF (how to fit, etc). The information is very important to understand the results.

Minor points

I think that there are several mistakes in writing in this manuscript (see below). Please check out the manuscript.

1) P23, Molecular dynamics simulation, “3ms” � “3 μs” ? It causes misunderstanding.

2) P3, amino-terminal domain()

3) P23, initial structure, “6TLE”�”6TLC”?

4) authors should define abbreviation of interleukin (IL)

5) P12, Figure 5H �5F.

6) P14, authors should add the descriptions on M0-M6 in caption of Figure 5.

7) P14, Figure 5B, the figure and letters are too small to understand.

**Have the authors made all data and (if applicable) computational code underlying the findings in their manuscript fully available?**

Reviewer #1: Yes

Reviewer #2: **No: **The authors could share the coordinates of the starting structures used for simulations and figures as also the raw data of the trajectories file. This is not a usual procedure, but this depends on the policy of the journal

Reviewer #3: Yes

PLOS authors have the option to publish the peer review history of their article (what does this mean?). If published, this will include your full peer review and any attached files.

Reviewer #1: No

Reviewer #2: No

Reviewer #3: No
---

## [Decision Letter · Decision Letter 1]

3 Sep 2022

Dear Ms. Zhao,

Thank you very much for submitting your manuscript "Allosteric Regulation in STAT3 Interdomains is Mediated by a Rigid Core: SH2 Domain Regulation by CCD in D170A Variant" for consideration at PLOS Computational Biology. As with all papers reviewed by the journal, your manuscript was reviewed by members of the editorial board and by several independent reviewers. The reviewers found the revised manuscript much improved; however, there are some points raised that require your attention (outlined in their comments). We are likely to accept this manuscript for publication, providing that you modify the manuscript according to the review recommendations.

Sincerely,

James Gallo

Academic Editor

PLOS Computational Biology

Nir Ben-Tal

Section Editor

PLOS Computational Biology

[LINK]

Reviewer's Responses to Questions

**Comments to the Authors:**

Reviewer #1: Thank you for carefully addressing the concerns raised by myself and the other reviewers. I have no further concerns.

Reviewer #2: The general quality of the revised version of the paper “Allosteric Regulation in STAT3 Interdomains is Mediated by a Rigid Core: SH2 Domain Regulation by CCD in D170A Variant” from Devin Matthews and co-workers is surely improved with respect to the previous version, and many concerns raised from the reviewers, mine included, have been faced and solved. However, the new data leaves unanswered two central questions. First, are the six simulations truly representative of the protein dynamics? And second, is the mechanism connecting the tilt of the α3 helix in the CCD and the dynamics of the pY+3 pocket in the SH2 domain rigorously demonstrated? Let me start with the second question. I think that figure S7, mainly the B panel (that above on the right, as letters are reported in the caption but not in the figure) clearly shows that this correlation does not exist. In all the states the tilt of the helix populates the same distribution (with a slight exception for the distribution in the M0 state), and I really don’t understand how the P-values reported in table S2 can reach so low numbers (e.g. 2.03E-306, between M2 and M3), how many freedom degrees have been considered? By the way, but this is a minor point, I definitively prefer to distinguish between p < 0.05 for statistically significant correlations and < 0.001 for highly significant ones, (What is the real difference between 2.03E-306 for M2/M3 and 0 for M1/M2?). In addition, also by looking at figure 4, it seems that the conformation of the α3 helix is the same in the five states. I think that the authors need to search for more robust pieces of evidence of the correlation between α3 helix and SH2 dynamics, otherwise, different explanations have to be proposed. The sentence “The rigidity of the α helices diminished the utility of PCA analysis”, is surely true for inter-domain PCA, but I think that PCA calculated on a sub-space (e.g the α3 helix in the CCD domain and the SH2 domain could enforce the proposed picture) could be helpful. Beyond this aspect, in my opinion, concerns remain on the efficacy of sampling. The authors write (page 9 line 276): “The long transition timescale between these states validates our kinetic clustering model, but also necessitates a “global” view of the trajectories in order to explain the full conformational dynamics of the protein". If long transition timescales characterize the system, the importance of each state is truely difficult to be predicted, as also cannot be ruled out that other states (not sampled in the six simulations) play a role. Furthermore, if stochastic transitions characterize the dynamics in the simulations, how can the authors exclude the possibility that wild-type will populate the M0, M1, and M3 states in further simulations? Probably for so complex system, enhanced sampling techniques could be preferable.

A few minor points remain:

1) In the answer to my question 20, in which I asked for SASA analysis, the authors answer that “we don’t have the facility to do it”, but SASA can be calculated with different free software without particular difficulties. I invite again the authors to calculate these data.

2) In the caption of figure 1 “there us” instead of “there is”

3) The figures in SI are recalled quite randomly.

4) Page 6 line 154 “The hypothesis was tested by comparing corresponding helical tilt at the CCD of different macro-states identified in Conformational differences in the pY+3 binding pocket correlate to the decreased binding affinity from wild type to D170A variant for the pY and pY+3 pockets.” The sentence is not clear to me.

5) Caption figure 4 indicates α21, whilst in the figure α27 is highlighted

Reviewer #3: The manuscript has been much improved.

This paper is an important contribution and I recommend that it be accepted for publication.

**Have the authors made all data and (if applicable) computational code underlying the findings in their manuscript fully available?**

Reviewer #1: Yes

Reviewer #2: Yes

Reviewer #3: Yes

PLOS authors have the option to publish the peer review history of their article (what does this mean?). If published, this will include your full peer review and any attached files.

Reviewer #1: No

Reviewer #2: No

Reviewer #3: No

Figure Files:

Data Requirements:

Reproducibility:

References:

---

## [Decision Letter · Decision Letter 2]

28 Sep 2022

Dear Ms. Zhao,

Thank you very much for submitting your manuscript "Allosteric Regulation in STAT3 Interdomains is Mediated by a Rigid Core: SH2 Domain Regulation by CCD in D170A Variant" for consideration at PLOS Computational Biology. As with all papers reviewed by the journal, your manuscript was reviewed by members of the editorial board and by several independent reviewers. 

Your second revision has improved the paper, yet there is still an issue raised by a reviewer concerning a correlation between the α-helical tilt and the dynamics of the binding site. Please address this issue in a revised manuscript. 

Sincerely,

James Gallo

Academic Editor

PLOS Computational Biology

Nir Ben-Tal

Section Editor

PLOS Computational Biology

[LINK]

Reviewer's Responses to Questions

**Comments to the Authors:**

Reviewer #2: In the previous revision, I had two main concerns. Concerning the first one, the authors have added a sentence of caution regarding the efficacy of the sampling; considering the complexity of the system, this can be considered sufficient.

On the contrary, practically nothing has been done to demonstrate a correlation between the movement of α3 helix in the CCD domain and the binding site on the SH2 domain. In my opinion, data reported in figure 3F and in figure S4 show that a correlation is absent. The authors wrote that “as example, small rotation of Ile252 (located at α3) could shift Gln511 (located at α21) because of the conserved the hydrogen network, which end up pushing away ILE569(located at α26) from β22.”. This suggests that probably a correlation exists but it does not necessarily involve the movements of the α3 helix (that is a global variable, probably less affected by small variations). To support their statement the authors refer to data in table S2. In this table now the degrees of freedom are reported. These numbers could be an overestimation of the real degrees of freedom; for example, it is unclear to me if the considered conformations have been sampled with an interval of time greater than the time decay of the autocorrelation of the helical tilt. If this is not the case, the considered conformations would not be completely independent (as a part of a trajectory), the real degrees of freedom could be significantly lower and, in turn, the p-values sensitively higher than those reported. In summarizing, I remain of the idea that a correlation between the α-helical tilt and the dynamics of the binding site is not demonstrated. As stated above, a correlation between some structural features in the CDD domain and the dynamics of the binding site in the SH2 domain could exist, but this correlation probably involves features more punctual than the helical tilt.

**Have the authors made all data and (if applicable) computational code underlying the findings in their manuscript fully available?**

Reviewer #2: Yes

PLOS authors have the option to publish the peer review history of their article (what does this mean?). If published, this will include your full peer review and any attached files.

Reviewer #2: No

Figure Files:

Data Requirements:

Reproducibility:

References:

---

## [Decision Letter · Decision Letter 3]

5 Dec 2022

Dear Ms. Zhao,

We are pleased to inform you that your manuscript 'Allosteric Regulation in STAT3 Interdomains is Mediated by a Rigid Core: SH2 Domain Regulation by CCD in D170A Variant' has been provisionally accepted for publication in PLOS Computational Biology.

Best regards,

James Gallo

Academic Editor

PLOS Computational Biology

Nir Ben-Tal

Section Editor

PLOS Computational Biology

Reviewer's Responses to Questions

**Comments to the Authors:**

Reviewer #2: The modified paper and the comments of the authors answer my questions and the paper is in my opinion ready for publication

**Have the authors made all data and (if applicable) computational code underlying the findings in their manuscript fully available?**

Reviewer #2: None

PLOS authors have the option to publish the peer review history of their article (what does this mean?). If published, this will include your full peer review and any attached files.

Reviewer #2: No

---

## [Editor Report · Acceptance letter]

15 Dec 2022

PCOMPBIOL-D-22-00920R3 

Allosteric Regulation in STAT3 Interdomains is Mediated by a Rigid Core: SH2 Domain Regulation by CCD in D170A Variant

Dear Dr Zhao,

I am pleased to inform you that your manuscript has been formally accepted for publication in PLOS Computational Biology. Your manuscript is now with our production department and you will be notified of the publication date in due course.

With kind regards,

Zsofi Zombor
